# Potentially inappropriate medication and attitudes of older adults towards deprescribing

**Alexandra B. Achterhof**[1,2], **Zsofia Rozsnyai**[1], **Emily Reeve**[3,4,5], **Katharina Tabea Jungo**[1,2], **Carmen Floriani**[1], **Rosalinde K. E. Poortvliet**[6], **Nicolas Rodondi**[1,7], **Jacobijn Gussekloo**[6,8], **Sven Streit**[1] *

1 Institute of Primary Health Care Bern (BIHAM), University of Bern, Bern, Switzerland, 2 Graduate School for Health Sciences, University of Bern, Bern, Switzerland, 3 Quality Use of Medicines and Pharmacy Research Centre, UniSA: Clinical and Health Sciences, University of South Australia, Adelaide, SA, Australia, 4 Geriatric Medicine Research, Dalhousie University and Nova Scotia Health Authority, Halifax, NS, Canada, 5 College of Pharmacy, Dalhousie University, Halifax, NS, Canada, 6 Department of Public Health and Primary Care, Leiden University Medical Center, Leiden, The Netherlands, 7 Department of General Internal Medicine, Inselspital, Bern University Hospital, University of Bern, Bern, Switzerland, 8 Department of Internal Medicine, section Gerontology and Geriatrics, Leiden University Medical Center, Leiden, The Netherlands

* sven.streit@biham.unibe.ch

**Data Availability Statement:** All data are available in the Supporting Information files.

**Funding:** This study was funded with a grant (PI Prof. S. Streit) by the Swiss Society of General

## Abstract

### Introduction

Multimorbidity and polypharmacy are current challenges when caring for the older population. Both have led to an increase of potentially inappropriate medication (PIM), illustrating the need to assess patients' attitudes towards deprescribing. We aimed to assess the prevalence of PIM use and whether this was associated with patient factors and willingness to deprescribe.

### Method

We analysed data from the LESS Study, a cross-sectional study on self-reported medication and on barriers and enablers towards the willingness to deprescribe (rPATD questionnaire). The survey was conducted among multimorbid ($\geq$3 chronic conditions) participants $\geq$70 years with polypharmacy ($\geq$5 long-term medications). A subset of the Beers 2019 criteria was applied for the assessment of medication appropriateness.

### Results

Data from 300 patients were analysed. The mean age was 79.1 years (SD 5.7). 53% had at least one PIM (men: 47.8%%, women: 60.4%%; p = 0.007). A higher number of medications was associated with PIM use (p = 0.002). We found high willingness to deprescribe in both participants with and without PIM. Willingness to deprescribe was not associated with PIM use (p = 0.25), nor number of PIMs (p = 0.81).

### Conclusion

The willingness of older adults with polypharmacy towards deprescribing was not associated with PIM use in this study. These results suggest that patients may not be aware if they

Internal Medicine (SGAIM). ER is support-ed by a NHMRC-ARC Dementia Research Development Fellowship.

**Competing interests:** The authors have declared that no competing interests exist.

are taking PIMs. This implies the need for raising patients' awareness about PIMs through education, especially in females, in order to implement deprescribing in daily practice.

## Background

An ageing population with multimorbidity ($\geq$3 chronic conditions) and polypharmacy ($\geq$5 long-term medications) poses a worldwide challenge to healthcare organisations, particularly in primary healthcare. As the prevalence of polypharmacy has increased due to high multimorbidity in especially the older population, potentially inappropriate medication (PIM) use has increased as well [1–3]. The single most important risk factor for PIM use is the number of prescribed medications [4]. Medications are considered 'potentially inappropriate' when its potential risk outweighs its clinical benefit in an individual [5]. Previous studies have reported a prevalence of PIMs between 40–80% [6–9]. Due to associated negative health consequences (e.g. reduced adherence and quality of life and increased risk of adverse drug reactions and hospitalizations), PIMs are an unnecessary burden to the older population [10–12].

Appropriate prescribing in the older population is challenging. First, older individuals have an increased risk of medication-related harm due to an age-related change in pharmacokinetics and -dynamics, a lower physiological reserve and drug-drug or drug-disease interactions [13–15]. Additionally, they are more susceptible to PIMs due to a lack of evidence regarding the benefits and harms of medications in multimorbid older adults and the frequently observed "prescribing cascade" where new medication is prescribed to treat a side effect of another medication [16]. Lastly, the application of single disease evidence-based guidelines to an individual with multimorbidity results in complex polypharmacy as they do not take into account potential drug- and disease-drug interactions [17,18].

The high prevalence and negative impact of PIMs, as well as the need to individualise therapy illustrates the importance of deprescribing in older individuals. Deprescribing is the process of withdrawal or dose reduction of inappropriate medications, supervised by a healthcare professional. This is endorsed by more recent guidelines, such as the NICE guidelines on multimorbidity and medication optimisation, that were developed to reduce polypharmacy and PIMs by recommending approaches on how to best manage and optimise pharmaceutical treatment in complex older adults [19,20].

Currently, deprescribing tools that assist physicians in detecting PIMs are increasingly being applied in daily practice. An example is the AGS Beers criteria, which is a globally used tool that lists PIMs that should be avoided in most older adults due to increased risk of harm or low/no benefit. Deprescribing can have a considerable positive impact on the health status and treatment burden of the older multimorbid population [21]. It may reduce adverse drug reactions, improve patients' quality of life and promote medication adherence [22–24]. Understanding patients' attitudes towards their medications and deprescribing can inform patient-centered care which is a key part of all clinical care [18].

Patients beliefs and attitudes towards deprescribing have increasingly been investigated [1,25–30], but whether these are correlated with appropriateness of their medications has not yet been determined. So far, quantitative research has mostly reported patients' and clinicians' attitudes towards deprescribing and investigated its relationship with patient-related factors (such as age). To date, the only medication-related factor that the revised Patients' Attitudes Towards Deprescribing questionnaire (rPATD) has been related with is the number of prescribed medications, with studies finding inconsistent results. It is not yet known how attitudes towards deprescribing may be related to the suitability of that individual for deprescribing (i.e.

whether they are taking a PIM). We hypothesized that patients who use PIMs experience more side effects than patients who do not use any PIMs, which in turn might affect their willingness to deprescribe.

In this study, we investigated whether there is an association of PIM use and willingness to deprescribe in older individuals and which factors influence patients' attitudes towards deprescribing. Second, we were interested to see how prevalent PIM was in a population of older patients with polypharmacy and which types of PIMs were most commonly used in men and women.

## Methods

### Design

The current study was nested in the LESS Study [31], which is a cross-sectional anonymous survey-study that evaluates the overall willingness to deprescribe and the barriers and enablers towards the willingness to deprescribe in older Swiss individuals with multimorbidity and polypharmacy. This manuscript reports the results of patients' attitudes towards deprescribing related to PIMs.

### Study population

Sixty-four general practitioners (GPs) from the German-speaking part of Switzerland recruited primary care patients for involvement in this study. All of them were located in different GP offices. Eligible patients were ≥70 years old with multimorbidity and polypharmacy. Multi-morbidity was defined as the presence of ≥3 chronic diseases, with chronic diseases being present for at least six consecutive months [32]. Polypharmacy was defined as the concurrent use of ≥5 long-term medications [33,34]. GPs were instructed to consecutively screen eligible patients and recruit 5 participants, reporting the number of patients screened, to reduce the risk of selection bias. The questionnaire was completed by a total of 306 patients, 6 of whom were excluded based on missing information about prescribed medication. Patients anonymously filled in the survey and handed it back to the practice nurse to limit the chance of social desirability bias.

### Questionnaire

For this study, we used data from 300 questionnaires on demographic status like age, gender, living situation, help with medication intake, involvement in medication self-management and education level. As for willingness to deprescribe, we used data from the revised Patients' Attitudes Towards Deprescribing questionnaire (rPATD). This is a validated and reliable tool that has been applied in multiple studies [1,35–38]. It contains 22 questions on a 5-point Likert scale, ranging from "strongly agree" to "strongly disagree" which relate to beliefs and attitudes about their medications and deprescribing [37]. The rPATD was translated into German as previously described [31].

### Medication appropriateness

In the present study, we used the self-reported list of prescribed medications and medication dosages for the assessment of PIMs. Self-reported medication is proven accurate and valid for long-term medication in the general population [39,40] and was chosen in this case to specifically focus on which medications patients report they take. The self-reported medication list was checked for inconsistencies (e.g. spelling errors) before analysis of PIMs was performed. In case of uncertainty regarding self-reported medication (e.g. due to poor or unreadable handwriting), in consultation with a GP researcher, the best applicable option was chosen

[41]. Next, each medication was coded according to the WHO ATC-coding system. For the assessment of medication appropriateness, a selection of the AGS Beers 2019 criteria was used [42]. The AGS Beers list is the most commonly used tool for assessment of PIMs worldwide [43]. Since data on medical conditions was limited in this study, we used only the criteria that were applicable without clinical information (52 of 97 criteria). A list of the included criteria is added in S1 Appendix. Criteria were excluded based on weak strength of evidence as defined in the AGS Beers list (n = 8) or lack of information (n = 37). Application of a subset of the Beers 2019 criteria is in line with previous studies that used subsets of the Beers criteria for assessing medication appropriateness [29,44–46].

## Willingness to deprescribe

Our main outcome was the willingness to deprescribe in relation to medication appropriateness. We therefore analysed data from the rPATD where patients were asked if they are satisfied with their current medications and if they are willing to deprescribe if their doctor said it was possible, along with 20 other questions grouped into four factors: involvement, burden, appropriateness and concerns about stopping, as described elsewhere [1].

## Ethics

Ethical approval for this study was obtained from the Ethics Committee of the Canton of Bern, Switzerland (Ref. 2017–02188). All patients provided written informed consent before participating in the study.

## Statistical analysis

Before the analysis, consistency checks were performed on the complete data set including the AGS Beers criteria and uncertainties were resolved by consensus of two researchers. Descriptive results were presented in frequencies, proportions, means and standard deviations (SD), and 95% confidence intervals (CI) were appropriate. Hypothesis testing for categorical variables was done using Chi-squared tests and simple linear regression for continuous variables when normally distributed. Patients with at least one PIM in their medication list where grouped to 'PIM yes', all others to 'PIM no' (exposure).

The individual scores (n = 22) of the rPATD showed a non-normal distribution. For the multivariate model we therefore dichotomized each of the 5-point Likert questions as well as factor scores according to the median as done previously [1]. Individual scores equal to or higher than the median were placed in the "high score" group, whereas scores below the median were placed in the "low score" group. In a multivariate model with different components of the rPATD as the outcome (satisfaction, willingness to stop, involvement, burden, appropriateness, concerns about stopping) [1], we calculated odds ratios (OR) and adjusted for age, gender and number of medications. To account for possible clustering of answers from patients from the same GP, we chose a mixed-effects model with the individual GP as random-effects. In a sensitivity analysis, we repeated the same models but with number of PIMs as the exposure instead of PIM yes vs. no. Significance level was set at <0.05. Data analyses were performed using STATA version 15.2 (Stata Corp, College Station, TX, USA).

## Results

For the overall analysis of polypharmacy levels and PIM, 300 participants were included, collectively taking approximately 2700 medications. Seventy-eight percent of all participants used 5–9 regular medications, with the remaining 22% using ≥10 medications (excessive

polypharmacy). Participants received on average 8 medications (SD 2.7). More than half of our sample (54%) received at least 1 PIM. The majority received 1 PIM (31.3%), 12.7% received 2 PIMs and 9.7% received ≥3 PIMs up to 7 PIMs. Gender distribution was approximately even, and participants had a mean age of 79.1 years (SD 5.7). The baseline characteristics of participants in each group (no PIM, 1 PIM and >1 PIM) are presented in Table 1. Age, living situation, medication self-management and education level did not significantly differ between participants receiving appropriate medication or PIM. We did, however, find an association of females having more PIM (p = 0.007) than men. Additionally, an association was found between a higher number of prescribed medications and PIMs (p = 0.002). Thus, patients receiving 10 or more medications (i.e. excessive polypharmacy) showed a significantly higher risk of taking PIMs (Table 1).

Fig 1 illustrates the proportion of participants who agreed to the individual questions about satisfaction with treatment and willingness to deprescribe and the proportion of participants with high factor scores stratified by PIM. The majority of participants were satisfied with their current medications (97.1% without PIM vs. 96.9% with PIM; p = 0.90) and were willing to have one or more of their medications deprescribed (74.3% without PIM, 79.9% with PIM; p = 0.25). From the four factor scores, we found more participants with PIM had high burden scores (61% vs. 49%; p = 0.029) and less had high concerns about stopping scores (53% vs. 65%; p = 0.034). The table in S2 Appendix provides more detail about the rPATD factors as shown in Fig 1. However, in the adjusted model the only association remaining was concerns about stopping which was significantly lower in patients with PIM compared to those without PIM (OR 0.55; 95%CI 0.33–0.92; p = 0.023). Moreover, this association disappeared in the sensitivity analysis where number of PIMs was the exposure instead of PIM yes vs. no (OR 0.86; 95%CI 0.69–1.09; p = 0.21).

The level of agreement to all individual rPATD questions for participants with PIM as compared to participants without PIM is presented in Table 2. There was a statistically significant difference in the proportion of participants who agreed with the question, "If my doctor recommended stopping a medicine I would feel that he/she was giving up on me" in participants taking ≥1 PIM compared to those without PIM (OR 0.49; 95%CI 0.29–0.82); p = 0.006). In the sensitivity analysis with number of PIMs as the exposure, the association became weaker (OR 0.80; 95%CI 0.62–1.02; p = 0.07).

**Table 1. Baseline characteristics of participants stratified by medication appropriateness.**

| Baseline characteristics | Overall n = 300 | No PIM n = 139 (46%) | 1 PIM n = 94 (31%) | >1 PIM n = 67 (22%) | p-value[a] |
|---|---|---|---|---|---|
| Female, n % | 139 (46) | 55 (40) | 42 (45) | 42 (63) | 0.007 |
| Age, mean (SD) | 79.1 (5.7) | 79.0 (5.5) | 78.9 (6.0) | 79.5 (6.0) | 0.61 |
| Living alone, n % | 100 (34) | 49 (36) | 27 (29) | 24 (36) | 0.48 |
| Self-management of medication, n % | 257 (86) | 120 (87) | 83 (88) | 54 (82) | 0.48 |
| Education level, n % | | | | | 0.33 |
| Basic education | 86 (29) | 34 (24) | 28 (30) | 24 (36) | |
| Apprenticeship | 146 (49) | 68 (49) | 49 (52) | 29 (43) | |
| Higher education | 68 (23) | 37 (27) | 17 (18) | 14 (21) | |
| Number of medicines, mean (SD) | 8.0 (2.7) | 7.4 (2.3) | 7.8 (2.4) | 9.4 (3.5) | <0.001 |
| 5–9 medicines | 233 (78) | 117 (84) | 74 (79) | 42 (63) | |
| ≥10 medicines | 67 (22) | 22 (16) | 20 (21) | 25 (37) | 0.002 |

Abbreviations: SD, standard deviation; PIM, potentially inappropriate medication.

[a]p-value is significant at <0.05.

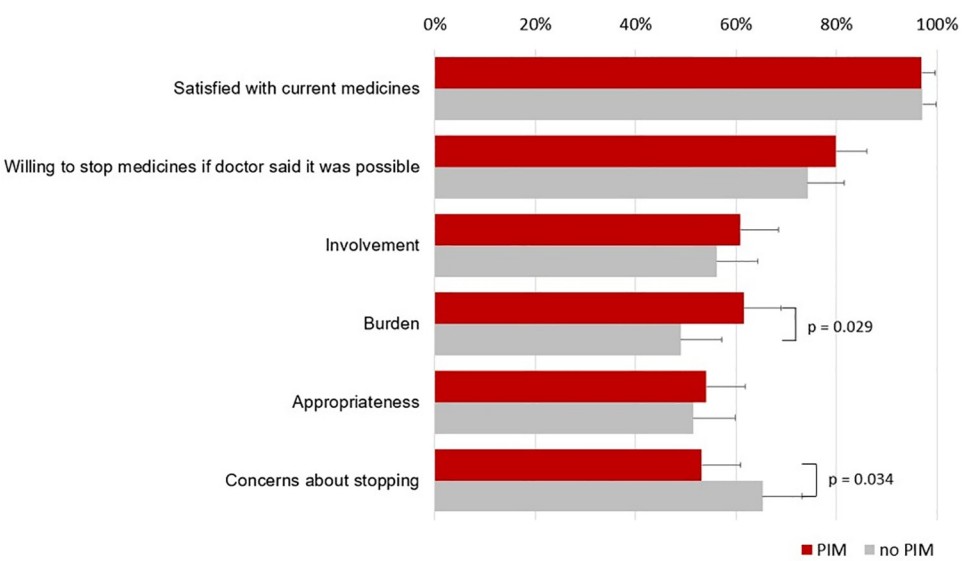

**Fig 1. Proportion of participants who agreed to the individual questions about satisfaction with treatment and willingness to deprescribe and the proportion of participants with high factor scores stratified by PIM.**
Involvement, burden, appropriateness and concerns about stopping are factor scores from the rPATD questionnaire [25]. Each of the four factors consisted of 5 questions of which the possible score ranged from 1–5. We grouped the answers of each patient to either 'yes' (if the factor score was higher than the median) or 'no' (if the factor score was lower than the median). We then calculate the proportion of patients answering "yes". Abbreviations: PIM, potentially inappropriate medication; rPATD, revised patients' attitudes towards deprescribing. p-value is significant at <0.05.

Fig 2 lists types of PIMs according to the 2019 AGS Beers criteria and their frequency by gender. We found proton pump inhibitors and benzodiazepines to be among the most frequent PIMs in our sample. Additionally, we found that certain PIMs differed by gender. Benzodiazepines (p<0.001), nonbenzodiazepines (p = 0.003), combinations of ≥3 CNS-active drugs (p = 0.001) and opioids in combination with benzodiazepines (p = 0.004) were significantly more frequent in females compared to males. Other drugs differed by gender as well including (peripheral alpha-1 blockers and estrogens).

# Discussion

## Summary

In this study, we found PIM to be prevalent (54%) in a consecutive sample of older patients with multimorbidity and polypharmacy in a primary care setting. PIM use was found to be higher in patients with more prescribed medications compared to less. Interestingly, females were more frequently prescribed a PIM, mostly benzodiazepines or other CNS-active drugs, than males. We observed no difference in patients' attitudes and willingness to deprescribe in patients with or without PIM. Our findings suggest that willingness to deprescribe is equally high in patients with and without PIM. There was also no difference in the adjusted analysis in burden, appropriateness or involvement factor scores, but participants taking PIM had lower concerns about stopping. This may therefore indicate that older adults with polypharmacy are not aware of whether they are taking potentially inappropriate medications or not. Therefore, efforts to increase awareness of the concept of PIM may be beneficial to shared-decision making about deprescribing in regular practice. Our study has also highlighted some areas that could be targeted, such as long term use of benzodiazepines in females.

**Table 2. Level of agreement to deprescribing in patients with PIM compared to patients without PIM.**

| rPATD questions[a] | Odds ratio[b] for PIM vs no PIM | 95% CI | p-value[c] |
|---|---|---|---|
| "Overall, I am satisfied with my current medicines" | 1.06 | 0.25–4.45 | 0.93 |
| "I like to be involved in making decisions about my medicines with my doctors" | 1.22 | 0.68–2.21 | 0.51 |
| "I have a good understanding of the reasons I was prescribed each of my medicines" | 1.34 | 0.79–2.29 | 0.28 |
| "I like to know as much as possible about my medicines" | 1.01 | 0.61–1.65 | 0.98 |
| "I always ask my doctor, pharmacist or other health care professional if there is something I don't understand about my medicine" | 1.13 | 0.68–1.89 | 0.63 |
| "I know exactly what medicines I am currently taking, and/or I keep an up to date list of my medicines" | 1.10 | 0.55–2.18 | 0.79 |
| "If my doctor said it was possible I would be willing to stop one or more of my regular medicines" | 1.60 | 0.87–2.94 | 0.13 |
| "I feel that I am taking a large number of medicines" | 1.37 | 0.81–2.30 | 0.24 |
| "Taking my medicines every day is very inconvenient" | 0.89 | 0.51–1.54 | 0.67 |
| "I spend a lot of money on my medicines" | 0.91 | 0.54–1.53 | 0.72 |
| "Sometimes I think I take too many medicines" | 1.21 | 0.73–2.01 | 0.46 |
| "I feel that my medicines are a burden to me" | 0.86 | 0.53–1.41 | 0.56 |
| "I would like to try stopping one of my medicines to see how I feel without it" | 0.85 | 0.51–1.40 | 0.51 |
| "I would like my doctor to reduce the dose of one or more of my medicines" | 1.18 | 0.72–1.92 | 0.52 |
| "I feel that I may be taking one or more medicines that I no longer need" | 1.47 | 0.91–2.38 | 0.12 |
| "I believe one or more of my medicines may be currently giving me side effects" | 1.14 | 0.68–1.92 | 0.61 |
| "I think one or more of my medicines may not be working | 0.94 | 0.13–7.08 | 0.95 |
| "I have had a bad experience when stopping a medicine before" | 0.60 | 0.35–1.03 | 0.06 |
| "I would be reluctant to stop a medicine that I had been taking for a long time" | 0.71 | 0.43–1.15 | 0.16 |
| "If one of my medicines was stopped I would be worried about missing out on future benefits" | 0.77 | 0.48–1.25 | 0.30 |
| "I get stressed whenever changes are made to my medicines" | 0.82 | 0.50–1.36 | 0.45 |
| "If my doctor recommended stopping a medicine I would feel that he/she was giving up on me" | 0.49 | 0.29–0.82 | 0.006 |

Abbreviations: rPATD, revised patients' attitudes towards deprescribing; PIM, potentially inappropriate medication; CI, confidence interval.

[a] from [25].

[b] Odds ratio is adjusted for age, sex, number of medicines and general practitioners.

[c] p-value is significat at <0.05.

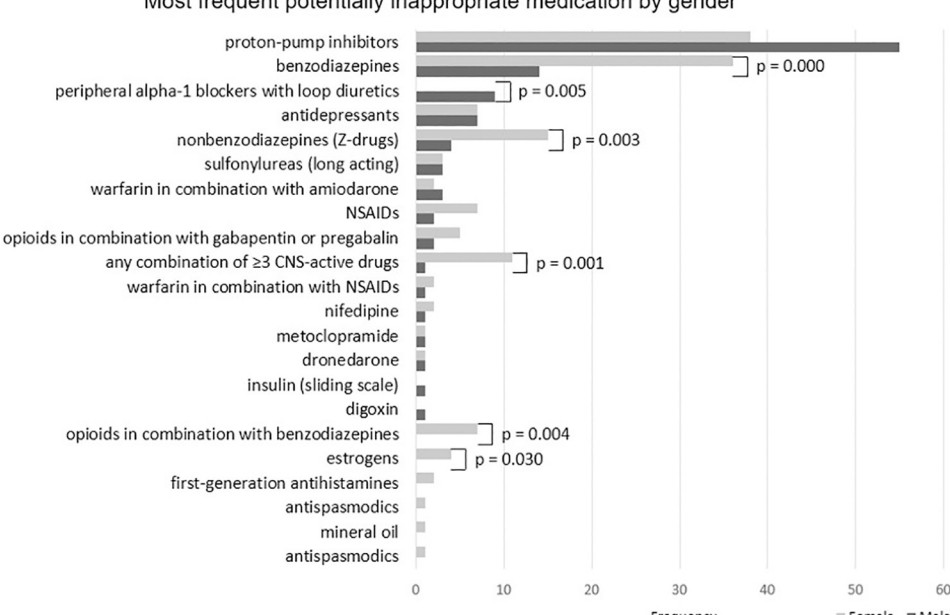

**Fig 2. Potentially inappropriate medication stratified by gender.** Abbreviations: NSAIDs, nonsteroidal anti-inflammatory drugs; CNS, central nervous system. p-value is significant at <0.05.

## Comparison with existing literature

The proportion of participants receiving at least one PIM matches previous studies from several countries worldwide, which generally report prevalence's between 40 and 80% [6–9,47]. We demonstrated that patients receiving 10 or more medications show a significantly higher risk of PIMs. This confirms findings from previous studies that the number of medications is the most important risk factor for PIMs [4,8,48]. Furthermore, we detected a correlation between gender and prevalence of PIMs. It has been previously reported that females receive a higher number of PIMs on average than males [7,47–50]. It has been suggested that this might be due to females being at a higher risk for developing multiple chronic conditions compared to males. This would imply that they are more susceptible to drug-drug and drug-disease interactions, which challenges appropriate prescribing [47]. Yet, the actual reason for this gender-difference is unknown. Similar to previous studies internationally [29,51], proton-pump inhibitors and benzodiazepines were the most common PIMs identified in our study population.

Interestingly, we found that patients' reported willingness to deprescribe is not related to PIM use according to the 2019 AGS Beers criteria. As investigated in a recent study on PIM and deprescribing interventions that explored factors associated with deprescribing refusal, likewise, PIM use was not associated with acceptance or refusal of deprescribing [29]. Previous qualitative studies on deprescribing, have reported that patients generally lack knowledge of the potential harms of medications and rely on the GP as a central and prominent figure in decision-making [52,53]. These findings suggest that older adults are not aware of whether their medications are appropriate or not. Reported willingness to deprescribe was equally high in our participants with and without PIM. Furthermore, very few of the individual factors had evidence to support a relationship with the use of PIMs, which stems from the fact that the study might not have been sufficiently powered to detect such differences. The factors 'burden' and 'concerns about stopping' were associated in the unadjusted analyses, but the burden

association was lost in the adjusted analyses (likely because of the confounding nature of PIMs being associated with number of medications). Therefore, it is still unclear if and how use of inappropriate medications influences attitudes and beliefs or vice versa. The overall high willingness of older adults with polypharmacy to deprescribe is promising for further implementation of deprescribing in primary healthcare and is in line with prior studies investigating patients' attitudes towards deprescribing [1,25–30].

## Limitations and strengths

We acknowledge several limitations in our study. First, we gathered information on prescribed medications through self-reported medication lists, which might have affected the completeness of the medication lists. The accuracy of self-reported medication data can vary with medication type and duration; with self-reporting generally being more accurate for long-term medications [39,40]. Specifically, certain medication categories (e.g. psychoanaleptics and analgesics) were previously found to be less reliably self-reported [40]. Therefore, by using self-reported medication lists in this study, our results may be an underestimation of the use of PIMs. Second, the prevalence of PIMs in our sample might be underestimated due to the limited generalizability of the Beers criteria, published by the American Geriatrics Society, as they are based on medications commonly prescribed in the United States. Additionally, the Beers criteria capture 'potentially' inappropriate medications and so the assessment of appropriateness is not individualised. Third, the recruitment of study participants by GPs could have introduced selection bias. However, since consecutive sampling was used for participant inclusion–as it was not possible to recruit a random sample–the risk of selection bias was minimised. Lastly, since our sample consisted of Swiss older patients, we do not know if our findings are generalizable to other populations. However, as other countries reported similar attitudes towards deprescribing (e.g. 88% willingness to deprescribe in Australia [1], 89% in Italy [25] and 92% in the USA [27]) this increases the confidence in our findings.

To the best of our knowledge, we were the first to investigate the relation between medication appropriateness and patients' willingness to deprescribe using validated tools. The Beers criteria is the most widely used tool for medication appropriateness and has proven to be accurate in the assessment of PIM [42,43]. The Beers 2019 version is updated according to the latest evidence and includes drug-drug interactions when assessing PIMs. Lastly, we used the rPATD questionnaire, which is validated and has been used internationally to assess willingness to deprescribe [37,54]. We followed international standards with independent forward and back translation to translate the rPATD into German.

## Implications for future practice and research

Although we did not detect an association of PIM and willingness to deprescribe, we did see positive trends of patients on PIM towards, for instance, the perception of having more side effects (14%) and taking medication they no longer need (47%). However, since willingness to deprescribe is equal, this implies that patients' willingness does not seem to be driven by knowledge that they are on a PIM. This again indicates the need to raise awareness about PIMs in older patients with polypharmacy, especially in females. Clinicians should be encouraged to regularly discuss deprescribing and the fact that the risks and benefits of medication use can change over time. Currently, patient education materials are increasingly being developed that will likely add to patients understanding of PIMs [55]. However, the group most at risk for PIM are vulnerable (oldest-) old patients [13–15] that demonstrate highly varying care wishes and needs, thereby challenging clinicians to provide appropriate care. Hence, in addition to our main finding–medication appropriateness being independent of patients'

willingness to have medication deprescribed–this pleads for an even bigger role of shared decision-making in the deprescribing process.

Future studies should further investigate the relationship between enabling factors of deprescribing and medication appropriateness and whether patients' attitudes and beliefs about medications may change with education [56]. Furthermore, they should focus on what patients consider inappropriate medication and which medications they would be willing to stop. Specific questions about patients' awareness about PIMs should be included in future research. As we found PIMs to be more prevalent in females compared to males, gender specific causes of PIM should be assessed in future studies. We also suggest focusing on specific classes and/or categories of medications in future research into PIM and willingness to deprescribe as this has not yet been explored and could be informative for translating into practice. Lastly, future studies should apply multiple PIM assessment tools, as well as comprehensive medication reviews determining actual appropriateness.

## Conclusion

We found PIM to be prevalent in the older population and patients to be generally willing to deprescribe. Patients' willingness to deprescribe was found to be irrespective of whether they were taking one or more PIMs. Female gender and increasing number of prescribed medications were positively associated with PIM use. Our results imply that it is necessary to raise awareness among older patients on PIMs, especially in females.

## Supporting information

**S1 File.**
(XLS)

**S1 Appendix. List of included Beers 2019 criteria.**
(TIF)

**S2 Appendix. Results of the rPATD factor scores.**
(TIF)

## Acknowledgments

The authors like to thank all GPs who recruited patients from their practice as well as all patients for their time and efforts in participating in this study. We also want to thank Prof. A. Chiolero for his contribution.

## Author Contributions

**Conceptualization:** Alexandra B. Achterhof, Zsofia Rozsnyai, Emily Reeve, Katharina Tabea Jungo, Carmen Floriani, Rosalinde K. E. Poortvliet, Jacobijn Gussekloo, Sven Streit.

**Data curation:** Alexandra B. Achterhof, Zsofia Rozsnyai.

**Formal analysis:** Alexandra B. Achterhof, Sven Streit.

**Funding acquisition:** Sven Streit.

**Investigation:** Alexandra B. Achterhof, Katharina Tabea Jungo.

**Methodology:** Alexandra B. Achterhof, Zsofia Rozsnyai, Katharina Tabea Jungo, Carmen Floriani, Rosalinde K. E. Poortvliet, Jacobijn Gussekloo, Sven Streit.

**Project administration:** Zsofia Rozsnyai, Sven Streit.

**Resources:** Emily Reeve, Sven Streit.

**Software:** Sven Streit.

**Supervision:** Zsofia Rozsnyai, Sven Streit.

**Validation:** Sven Streit.

**Visualization:** Alexandra B. Achterhof.

**Writing – original draft:** Alexandra B. Achterhof, Zsofia Rozsnyai.

**Writing – review & editing:** Emily Reeve, Katharina Tabea Jungo, Carmen Floriani, Rosalinde K. E. Poortvliet, Nicolas Rodondi, Jacobijn Gussekloo, Sven Streit.

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
