## [Decision Letter · Decision Letter 0]

14 Jul 2020

PONE-D-20-16955

Potentially inappropriate medication and attitudes of older adults towards deprescribing

PLOS ONE

Dear Dr. Streit,

Thank you for submitting your manuscript to PLOS ONE. After careful consideration, we feel that it has merit but does not fully meet PLOS ONE’s publication criteria as it currently stands. Therefore, we invite you to submit a revised version of the manuscript that addresses the points raised during the review process.

We look forward to receiving your revised manuscript.

Kind regards,

Christine Leong, Pharm. D.

Academic Editor

PLOS ONE

Journal Requirements:

"This study was sponsored with a grant (PI Prof. S. Streit) from the Swiss Society of General Internal

Medicine (SGAIM) Foundation. 

ER is supported by a NHMRC-ARC Dementia Research Development Fellowship."

"This study was funded with a grant (PI Prof. S. Streit) by the Swiss Society of General Internal Medicine (SGAIM). "

Reviewers' comments:

Reviewer's Responses to Questions

**Comments to the Author**

1. Is the manuscript technically sound, and do the data support the conclusions?

Reviewer #1: Yes

Reviewer #2: Yes

2. Has the statistical analysis been performed appropriately and rigorously? 

Reviewer #1: I Don't Know

Reviewer #2: Yes

3. Have the authors made all data underlying the findings in their manuscript fully available?

Reviewer #1: Yes

Reviewer #2: No

4. Is the manuscript presented in an intelligible fashion and written in standard English?

Reviewer #1: No

Reviewer #2: Yes

5. Review Comments to the Author

Reviewer #1: General Comment: From the way the article is written, these authors appear to have assumed that patients are likely to be aware of the PIMs that they are taking. I am very surprised by this assumption. My surprise is compounded when I read this statement in the article's "Comparison with existing literature" section: "This is in line with previous qualitative studies on deprescribing, showing that patients generally lack understanding of potential harms of medicines and showing the GP as a central and prominent figure in decision-making when it comes to deprescribing." So they evidently knew of this to begin with. Because of this, I struggle with their overall framing of the paper.

Nonetheless, the paper makes a useful, modest contribution. They show that people are generally willing to consider deprescribing, and that patients need to be better informed about their medications. They also show that the PIM situation applies more to women than men.

I also think that their hypothesis could in fact be investigated with a revised design, perhaps along the following lines. First, identify a population of older patients who are on PIMs. Second, tell them they are on a PIM. Third, propose deprescribing. Record the response. Then explain the PIM, then propose again and record the response, and compare.

Specific Points to Address:

1. Method section of abstract. Statement should read "...towards the willingness to deprescribe..."

2. First sentence of the "Study population" section is incoherent.

3. Sixth line under "Medication appropriateness", parenthesis should be "poor or unreadable"

4. Same line, "...in team with...": replace "team" with "consultation" of "collaboration".

5. Results section, near top of page 9, "feeling less given up [on] by their physician..." -- ie insert "on"

6. First line of conclusion should be "...generally willing to deprescribe."

Reviewer #2: Thank you for the opportunity to review this manuscript, which reports a study of attitudes towards deprescribing and potentially inappropriate medicines in older adults with multimorbidity and polypharmacy. This paper is well written, and investigates an important, worthwhile and novel question. I have a number of minor comments and requests for clarifications that I hope will improve this already high quality paper.

Background

1. For the sentence "It can reduce the number of side effects, improve patients’ quality of life and promote medication adherence." - this is logical and supported by the evidence for polypharmacy/PIMs' impact on these outcomes. However perhaps any studies that have shown deprescribing impacts on these could be referenced, or the sentence changed to "It may reduce...".

2. The reference to assessment of PIM and deprescribing as "cornerstones" of primary healthcare could be revised. Although their importance may be well recognised, as you suggest their implementation is not optimal.

Methods

3. The Study population section refers to 64 GPs recruiting. Were these all from difference general practices/primary care centres?

4. The same sentence refers to "the German part of Switzerland". Is this the most appropriate description, or would "German-speaking part" be better instead?

5. The same section states "GPs were instructed to consecutively screen and select all eligible patients in a defined timeframe (e.g. 2 weeks)". Can you please clarify was this period the same for all GPs or how was the timeframe defined on an individual basis e.g. was this until a certain number of patients had accrued?

6. In the Medication appropriateness section, there is a sentence on the accuracy of self-report medication. While this is generally true, there are cases from the two studies referenced (and the literature as a whole) of medications with poorer agreement, such as psycholeptics and analgesics. This has implications given these medications feature often in the Beers criteria. I feel this should be expanded on here, and in the study limitations section.

7. It would be helpful to provide a list of the included 52 criteria in the appendix. Also, the "description of the method for counting of medicines" does not actually seem to be included in Appendix 1 at the moment.

Results

8. The sentence "We did, however, find an association of females having more PIM (p=0.007) than men with increasing numbers of PIM." is somewhat unclear and could benefit from rephrasing.

9. Again, reference to "...patients with PIM feeling less given up by their physician" is a little unclear. Perhaps something such as the following may be clearer: "...patients with PIM agreeing less that they felt their physician was giving up on them".

Discussion

10. The Discussion states "Furthermore, very few of the individual questions and factor scores were related with PIM use." It may be worth considering if the study was sufficiently powered to detect such differences. This could be acknowledged by rephrasing that very few of the individual questions and factor scores had evidence to support a relationship with PIM use.

11. The sentence beginning "Patients seem to not..." could be rephrased to say "Our findings suggest that patients seem to not...", just to clarify that this wasn't specifically investigated in this study.

Table 1.

12. I would suggest relabeling the <9 medicines categories to 5-9 medicines to re-emphasise that all patients were on at least 5 medicines.

Appendix 1

13. In the table titled "Results of the rPATD factor scores", it's unclear exactly what the % in the PIM/no PIM columns refer to, and the medians/IQRs. Could these be elaborated on in the table title, or in the table legend?

6. PLOS authors have the option to publish the peer review history of their article (what does this mean?). If published, this will include your full peer review and any attached files.

Reviewer #1: **Yes: **James Conklin

Reviewer #2: **Yes: **Frank Moriarty

---

## [Author Response · Author response to Decision Letter 0]

31 Aug 2020

Dear Dr Leong, dear PLOS ONE Editorial Board,

We were invited to submit a revised version of our manuscript: 

“Potentially inappropriate medication use and attitudes of older adults towards deprescribing”

We are very pleased to accept the opportunity to revise and resubmit a revised version of this manuscript and would like to thank you for this chance. 

As both reviewers’ comments and suggestions were very valuable, we were happy to im-plement them and feel that they have improved the overall quality of the manuscript. 

We wrote a response to the reviewers that is attached below, answering every comment point-by-point. With that, we hope to have sufficiently answered and implemented all ques-tions, suggestions and comments. 

We dearly hope to have made the necessary adaptations for convincing you to accept our submission. Again, we want to express our gratitude for the opportunity to improve our manuscript with the relevant and valuable input from the reviewers. 

Yours sincerely,

Sven Streit 

Journal Requirements:

1. Please ensure that your manuscript meets PLOS ONE's style requirements, includ-ing those for file naming. The PLOS ONE style templates can be found at

Response: Thank you for this reminder. We adapted our manuscript accordingly. 

- We changed 

o all headings of major sections to bold type 18pt font 

o all sub-sections of major sections into bold type 16pt font

o the citing figures from “Figure 1” to Fig 1 

o the referencing from . (xx) to [xx]. 

- We put tables directly after the paragraph in which they are first cited 

- We inserted figure captions directly after the paragraph in which they are first cited

2. Thank you for stating the following in the Acknowledgments Section of your man-uscript:

"This study was sponsored with a grant (PI Prof. S. Streit) from the Swiss Society of General Internal Medicine (SGAIM) Foundation. 

ER is supported by a NHMRC-ARC Dementia Research Development Fellowship."

We note that you have provided funding information that is not currently declared in your Funding Statement. However, funding information should not appear in the Acknowledgments section or other areas of your manuscript. We will only publish funding information present in the Funding Statement section of the online submis-sion form.

Please remove any funding-related text from the manuscript and let us know how you would like to update your Funding Statement. Currently, your Funding Statement reads as follows: "This study was funded with a grant (PI Prof. S. Streit) by the Swiss Society of General Internal Medicine (SGAIM). "

Response: We removed both sentences from the acknowledgement section of the manuscript. 

We would like to update the Funding statement to: “This study was funded with a grant (PI Prof. S. Streit) by the Swiss Society of General Internal Medicine (SGAIM). ER is support-ed by a NHMRC-ARC Dementia Research Development Fellowship.” 

 

Reviewer 1

Response: We thank the reviewer for the comments and suggestions on our paper. We have implemented them in our manuscript, as described in the point-by-point response below.

1. From the way the article is written, these authors appear to have assumed that pa-tients are likely to be aware of the PIMs that they are taking. I am very surprised by this assumption. My surprise is compounded when I read this statement in the arti-cle's "Comparison with existing literature" section: "This is in line with previous qualitative studies on deprescribing, showing that patients generally lack understand-ing of potential harms of medicines and showing the GP as a central and prominent figure in decision-making when it comes to deprescribing." So they evidently knew of this to begin with. Because of this, I struggle with their overall framing of the paper.

Response: Thank you for this comment. We have revised the manuscript to make it more explicit about our hypothesis (page 4). We hypothesized that people with PIM are experienc-ing more side effects, especially when using multiple PIMs simultaneously. Therefore, we thought that this could positively influence their willingness to deprescribe, as we can imag-ine that having many or worse side effects can drive the will to stop medication. So we did not necessarily assume that patients are likely to be aware of the PIMs that they are taking, but we assumed that, indirectly, (the number of) PIMs would influence patients’ willingness to have medication deprescribed due to a presumably higher number of side effects or severe side effects. We have added the following sentence to the manuscript to briefly summarize this: “We hypothesized that patients who use PIMs, irrespective of whether the medication they use are potentially inappropriate, experience more side effects than patients who do not use any PIMs, which in turn might affect their willingness to deprescribe” (page 4). 

2. I also think that their hypothesis could in fact be investigated with a revised design, perhaps along the following lines. First, identify a population of older patients who are on PIMs. Second, tell them they are on a PIM. Third, propose deprescribing. Rec-ord the response. Then explain the PIM, then propose again and record the response, and compare.

Response: Thank you for your comment. We agree that the design you mentioned could be used to investigate the same hypothesis that we addressed in our manuscript. We chose to conduct a cross-sectional study since we could use data collected during the LESS-study as mentioned in our methods section. We will keep your suggestions in mind when working on future research projects to further examine the association between the use of potentially in-appropriate medications and patients’ willingness to deprescribe.

3. Specific Points to Address:

1) Method section of abstract. Statement should read "...towards the willingness to deprescribe..."

2) First sentence of the "Study population" section is incoherent.

3) Sixth line under "Medication appropriateness", parenthesis should be "poor or unreadable"

4) Same line, "...in team with...": replace "team" with "consultation" of "collabora-tion".

5) Results section, near top of page 9, "feeling less given up [on] by their physician..." -- ie insert "on"

6) First line of conclusion should be "...generally willing to deprescribe."

Response: Thank you for highlighting these typos. We corrected all of them. 

 

Reviewer 2: 

Response: We thank the reviewer for the helpful comments on our paper. We address the comments individually in our point-by-point response below.

1. Background: For the sentence "It can reduce the number of side effects, improve patients’ quality of life and promote medication adherence." - this is logical and sup-ported by the evidence for polypharmacy/PIMs' impact on these outcomes. However perhaps any studies that have shown deprescribing impacts on these could be refer-enced, or the sentence changed to "It may reduce...".

Response: Thank you for highlighting this. As suggested, we have changed the sentence to “It may reduce…” (page 4). 

2. Background: The reference to assessment of PIM and deprescribing as "corner-stones" of primary healthcare could be revised. Although their importance may be well recognised, as you suggest their implementation is not optimal.

Response: Thank you for your comment. We agree with it, and thus we removed this part. 

3. Methods: The Study population section refers to 64 GPs recruiting. Were these all from difference general practices/primary care centres?

Response: The 64 GPs that recruited patients for the LESS study were all practicing in dif-ferent general practices.

4. Methods: The same sentence refers to "the German part of Switzerland". Is this the most appropriate description, or would "German-speaking part" be better in-stead?

Response: We changed it into “the German-speaking part of Switzerland” (page 5).

5. Methods: The same section states "GPs were instructed to consecutively screen and select all eligible patients in a defined timeframe (e.g. 2 weeks)". Can you please clarify was this period the same for all GPs or how was the timeframe defined on an individ-ual basis e.g. was this until a certain number of patients had accured?

Response: All GPs were asked to consecutively recruit 5 study participants and to document the screening process of all eligible participants. We have adapted the text in the manuscript accordingly to reflect the information from our study protocol because the mentioning of "e.g. 2 weeks" was an example if a GP chose to define his time frame to this lenght knowing the population they take care for. The text now reads: “GPs were instructed to consecutively screen 5 eligible patients and to report the number of patients screened throughout the screen-ing process, to reduce the risk of selection bias.” (page 5).

6. Methods: In the Medication appropriateness section, there is a sentence on the ac-curacy of self-report medication. While this is generally true, there are cases from the two studies referenced (and the literature as a whole) of medications with poorer agreement, such as psycholeptics and analgesics. This has implications given these medications feature often in the Beers criteria. I feel this should be expanded on here, and in the study limitations section.

Response: Thank you for your comment. We do believe you address a very relevant topic and want to elaborate on that. It is indeed mentioned in some of the literature that specific drug categories are poorer self-reported. We are very much aware of the limitations that ac-company the use of self-reported data on medication intake. We have revised the limitation section of the manuscript accordingly (page 12 and 13).

7. Methods: It would be helpful to provide a list of the included 52 criteria in the ap-pendix. Also, the "description of the method for counting of medicines" does not ac-tually seem to be included in Appendix 1 at the moment.

Response: Thank you for your comment. We agree that it might be useful for the reader to have a list of included criteria and therefore, we added a list of the 52 included Beers criteria in Appendix 1. We removed the description of the method for counting of medicines from the appendix. 

8. Results: The sentence "We did, however, find an association of females having more PIM (p=0.007) than men with increasing numbers of PIM." is somewhat un-clear and could benefit from rephrasing.

Response: We have changed the sentence to: “We did, however, find an association of fe-males using more PIMs (p=0.007) than men, as well as females using more PIMs simulta-neously (page 8).

9. Results: Again, reference to "...patients with PIM feeling less given up by their physician" is a little unclear. Perhaps something such as the following may be clearer: "...patients with PIM agreeing less that they felt their physician was giving up on them".

Response: Thank you for your comment. We decided to revise the entire sentence and re-moved the part of the sentence that was unclear. 

10. Discussion: The Discussion states "Furthermore, very few of the individual ques-tions and factor scores were related with PIM use." It may be worth considering if the study was sufficiently powered to detect such differences. This could be acknowl-edged by rephrasing that very few of the individual questions and factor scores had evidence to support a relationship with PIM use.

Response: Thank you for this suggestion. We have replaced the sentence by: “very few of the individual factors had evidence to support a relationship with the use of PIMs, which stems from the fact that the study might not have been sufficiently powered to detect such differences” (page 11). 

11. Discussion: The sentence beginning "Patients seem to not..." could be rephrased to say "Our findings suggest that patients seem to not...", just to clarify that this wasn't specifically investigated in this study.

Response: We have decided to remove this sentence from the discussion, so it is no longer necessary to make an adjustment. 

12. Table 1: I would suggest relabelling the <9 medicines categories to 5-9 medicines to re-emphasise that all patients were on at least 5 medicines.

Response: We have adapted the manuscript text accordingly. 

13. Appendix 1: In the table titled "Results of the rPATD factor scores", it's unclear exactly what the % in the PIM/no PIM columns refer to, and the medians/IQRs. Could these be elaborated on in the table title, or in the table legend?

Response: Thank you for highlighting this. We have added further explanations to the figure 1 legend. We have added the following information to the figure legend: “Involvement, bur-den, appropriateness and concerns about stopping are factor scores from the rPATD ques-tionnaire. (39) Each of the four factors consisted of 5 questions of which the possible score ranged from 1-5. We grouped the answers of each patient to either ‘yes’ (if the factor score was higher than the median) or ‘no’ (if the factor score was lower than the median). We then calculate the proportion of patients answering “yes”.” (page 9).

---

## [Decision Letter · Decision Letter 1]

22 Sep 2020

PONE-D-20-16955R1

Potentially inappropriate medication and attitudes of older adults towards deprescribing

PLOS ONE

Dear Dr. Streit,

Thank you for submitting your manuscript to PLOS ONE. After careful consideration, we feel that it has merit but does not fully meet PLOS ONE’s publication criteria as it currently stands. Therefore, we invite you to submit a revised version of the manuscript that addresses the points raised during the review process.

We look forward to receiving your revised manuscript.

Kind regards,

Christine Leong, Pharm. D.

Academic Editor

PLOS ONE

Reviewers' comments:

Reviewer's Responses to Questions

**Comments to the Author**

1. If the authors have adequately addressed your comments raised in a previous round of review and you feel that this manuscript is now acceptable for publication, you may indicate that here to bypass the “Comments to the Author” section, enter your conflict of interest statement in the “Confidential to Editor” section, and submit your "Accept" recommendation.

Reviewer #2: (No Response)

2. Is the manuscript technically sound, and do the data support the conclusions?

Reviewer #2: Yes

3. Has the statistical analysis been performed appropriately and rigorously? 

Reviewer #2: Yes

4. Have the authors made all data underlying the findings in their manuscript fully available?

Reviewer #2: Yes

5. Is the manuscript presented in an intelligible fashion and written in standard English?

Reviewer #2: Yes

6. Review Comments to the Author

Reviewer #2: Thank you for the responses to my review. I feel they have addressed the majority of issues.

1. Thank you for the clarification that each GP was in a different practice/clinic. This should be added to the Methods section.

2. Regarding screening, the newly inserted sentence says GPs were instructed to consecutively screen 5 eligible patients. Should this be "to consecutively screen eligible patients and recruit 5 participants, reporting the number of patients screened..."?

3. The new legend that was intended for Figure 1 appears to have been inserted below the Figure 2 title in error.

4. Two revisions to the manuscript mentioned in the response letter have not actually been included in the manuscript (they have been inserted and then deleted in the tracked changes version). These related to hypothesising patients using PIMs were more likely to experience side effects in the Introduction, and that the study may not have been powered for examining individual questions in the Limitations. I feel these are both important and warrant inclusion.

5. The proposed addition relating to the hypothesis may warrant rephrasing as it is contradictory as is : "We hypothesized that

patients who use PIMs, irrespective of whether the medication they use are potentially inappropriate, experience...". Should the part beginning irrespective refer to the number of medications they use, or simply be removed?

7. PLOS authors have the option to publish the peer review history of their article (what does this mean?). If published, this will include your full peer review and any attached files.

Reviewer #2: **Yes: **Frank Moriarty

---

## [Author Response · Author response to Decision Letter 1]

25 Sep 2020

Recipient

Christine Leong, Pharm. D.

Academic Editor

PLOS ONE

Revision of manuscript ID: PONE-D-20-16955

Dear Dr Leong, dear PLOS ONE Editorial Board,

We were invited to submit a revised version of our manuscript: 

“Potentially inappropriate medication use and attitudes of older adults towards deprescribing”

We accepted the opportunity to revise and resubmit a revised version of this manuscript and we would like to thank you for this opportunity. 

We wrote a response to the reviewer’s comments that is attached below, answering every comment point-by-point. With that, we hope to have sufficiently answered and implemented all questions, suggestions and comments. 

We dearly hope to have made the necessary adaptations for convincing you to accept our revised submission. Again, we want to express our gratitude for the opportunity to improve our manuscript with the relevant and valuable input from the reviewer. 

Yours sincerely,

Prof. Sven Streit, MD, MSc, PhD

Professor in Primary Care, Head of Interprofessional Primary Care

Institute of Primary Health Care Bern (BIHAM)

University of Bern, Mittelstrasse 43

3012 Bern, Switzerland

Tel +41 31 631 58 75

Email: sven.streit@biham.unibe.ch

Point-by-point response to reviewers' comments:

1. If the authors have adequately addressed your comments raised in a previous round of review and you feel that this manuscript is now acceptable for publication, you may indicate that here to bypass the “Comments to the Author” section, enter your conflict of interest statement in the “Confidential to Editor” section, and submit your "Accept" recommendation.

Reviewer #2: (No Response)

Response: We would like to thank the reviewer for his positive feedback on our first revision. We have address the additional comments in the point-by-point response below. 

2. Is the manuscript technically sound, and do the data support the conclusions?

The manuscript must describe a technically sound piece of scientific research with data that supports the conclusions. Experiments must have been conducted rigorously, with appropri-ate controls, replication, and sample sizes. The conclusions must be drawn appropriately based on the data presented. 

Reviewer #2: Yes

Response: Thank you for this comment. We are happy to read that we addressed the com-ments in the first revision adequately. 

3. Has the statistical analysis been performed appropriately and rigorously?

Reviewer #2: Yes

Response: We would like to thank the reviewer for this assessment.

4. Have the authors made all data underlying the findings in their manuscript fully available?

The PLOS Data policy requires authors to make all data underlying the findings described in their manuscript fully available without restriction, with rare exception (please refer to the Data Availability Statement in the manuscript PDF file). The data should be provided as part of the manuscript or its supporting information, or deposited to a public repository. For ex-ample, in addition to summary statistics, the data points behind means, medians and variance measures should be available. If there are restrictions on publicly sharing data—e.g. partici-pant privacy or use of data from a third party—those must be specified.

Reviewer #2: Yes

Response: Thank you. 

5. Is the manuscript presented in an intelligible fashion and written in standard English?

Reviewer #2: Yes

Response: Thank you. 

6. Review Comments to the Author

Please use the space provided to explain your answers to the questions above. You may also include additional comments for the author, including concerns about dual publication, re-search ethics, or publication ethics. (Please upload your review as an attachment if it exceeds 20,000 characters)

Reviewer #2: Thank you for the responses to my review. I feel they have addressed the ma-jority of issues.

Response: We would like to thank the review for this assessment. The new comments are addressed in the point-by-point response below. 

1. Thank you for the clarification that each GP was in a different practice/clinic. This should be added to the Methods section.

Response: Thank you. We have added this information to the methods section (p. 5, l. 2). The text now reads: “All of them were located in different GP offices.” 

2. Regarding screening, the newly inserted sentence says GPs were instructed to consecu-tively screen 5 eligible patients. Should this be "to consecutively screen eligible patients and recruit 5 participants, reporting the number of patients screened..."?

Response: Thank you for highlighting this error. We have replaced the sentence as suggest-ed. The text now reads: “GPs were instructed to consecutively screen eligible patients and recruit 5 participants, reporting the number of patients screened, to reduce the risk of selec-tion bias” (p. 6, l. 6-7). 

3. The new legend that was intended for Figure 1 appears to have been inserted below the Figure 2 title in error.

Response: Thank you. We have moved the figure legend. 

4. Two revisions to the manuscript mentioned in the response letter have not actually been included in the manuscript (they have been inserted and then deleted in the tracked changes version). These related to hypothesising patients using PIMs were more likely to experience side effects in the Introduction, and that the study may not have been powered for examining individual questions in the Limitations. I feel these are both important and warrant inclusion.

Response: Thank you for highlighting this shortcoming. We have inserted these two revision in the manuscript text.

1) “We hypothesized that patients who use PIMs experience more side effects than patients who do not use any PIMs, which in turn might affect their willingness to deprescribe” (p. 4, l. 15-17). (this sentence was adapted based on the comment below) 

2) “Furthermore, very few of the individual factors had evidence to support a relationship with the use of PIMs, which stems from the fact that the study might not have been suffi-ciently powered to detect such differences.” (p. 11, l. 11-13) 

5. The proposed addition relating to the hypothesis may warrant rephrasing as it is contradic-tory as is : "We hypothesized that patients who use PIMs, irrespective of whether the medi-cation they use are potentially inappropriate, experience...". Should the part beginning irre-spective refer to the number of medications they use, or simply be removed?

Response: Thank you. We have removed the part “irrespective of whether the medication they use are potentially inappropriate” from the manuscript text. 

7. PLOS authors have the option to publish the peer review history of their article (what does this mean?). If published, this will include your full peer review and any attached files.

Response: I agree

---

## [Editor Report · Decision Letter 2]

28 Sep 2020

Potentially inappropriate medication and attitudes of older adults towards deprescribing

PONE-D-20-16955R2

Dear Dr. Streit,

We’re pleased to inform you that your manuscript has been judged scientifically suitable for publication and will be formally accepted for publication once it meets all outstanding technical requirements.

Kind regards,

Christine Leong, Pharm. D.

Academic Editor

PLOS ONE

---

## [Editor Report · Acceptance letter]

29 Sep 2020

PONE-D-20-16955R2 

Potentially inappropriate medication and attitudes of older adults towards deprescribing 

Dear Dr. Streit:

I'm pleased to inform you that your manuscript has been deemed suitable for publication in PLOS ONE. Congratulations! Your manuscript is now with our production department. 

Kind regards, 

on behalf of

Dr. Christine Leong 

Academic Editor

PLOS ONE